

# Common framework and quadratic Bethe equations for rational Gaudin magnets in arbitrarily oriented magnetic fields

**Alexandre Faribault**[1*] **and Hugo Tschirhart**[1,2]

**1** Groupe de Physique Statistique, Institut Jean Lamour (CNRS UMR 7198),
Université de Lorraine Nancy, B.P. 70239, F-54506 Vandoeuvre-lès-Nancy Cedex, France.
**2** Applied Mathematics Research Center, Coventry University, Coventry, England.

* alexandre.faribault@univ-lorraine.fr

## Abstract

In this work we demonstrate a simple way to implement the quantum inverse scattering method to find eigenstates of spin-1/2 XXX Gaudin magnets in an arbitrarily oriented magnetic field. The procedure differs vastly from the most natural approach which would be to simply orient the spin quantisation axis in the same direction as the magnetic field through an appropriate rotation.

Instead, we define a modified realisation of the rational Gaudin algebra and use the quantum inverse scattering method which allows us, within a slightly modified implementation, to build an algebraic Bethe ansatz using the same unrotated reference state (pseudovacuum) for any external field. This common framework allows us to easily write determinant expressions for certain scalar products which would be highly non-trivial in the rotated system approach.



# 1  Introduction

The rational Richardson-Gaudin [1–4] models define a class of integrable quantum spin models in which the coupling between spins is isotropic. The conserved charges are a set of isotropic central spin models centered around each of the different spins in the system. Their integrability and our capacity to find their exact eigenstates in the framework of the Algebraic Bethe Ansatz (ABA), is unaffected by the addition of an external magnetic field which simply modifies each conserved charge by adding a Zeeman term $\vec{B} \cdot \vec{S}_i$ coupling the external field exclusively to the corresponding spin $\vec{S}_i$. The isotropy of the coupling terms leads to a natural formulation of the ABA, in the eigenbasis formed by the tensor product of the eigenstates of each spin's Zeeman term $\vec{B} \cdot \vec{S}_i$. However, the consequence is that each possible orientation of the magnetic field would lead to eigenstates which are built explicitly in their own rotated basis.

In this work, we generalise the application of the Quantum Inverse Scattering Method (QISM) approach [5] in order to include in-plane components of an arbitrarily oriented magnetic field, but we do so by using a natural generalisation of the usual representation of the underlying Generalised Gaudin Algebra (GGA) [6]. Although the representation is simple to define, it lacks a highest-weight state which would serve as the reference state in the ABA. However, despite this fact, we show that a common pseudovacuum can actually be used to define a common framework for arbitrarily oriented fields, leading to a construction which is as closely related to the traditional ABA as can be.

In the next section we will first review the basic QISM and the typical requirements of the ABA, in the specific context of the rational family of Richardson-Gaudin models. Section 3 will then describe the specific details of the spin-1/2 realisation of the Gaudin magnets, and then define the alternative realisation of the GGA which will be used in the rest of this work.

In the fourth section we will demonstrate that, by relaxing a usual ABA constraint on the pseudovacuum, one can still apply the QISM and find proper Bethe equations for this system, for which, working with a common reference state, defines this common framework leading to similarly-built representations of the eigenstates for any magnetic field.

In section 5, we demonstrate that a change of variable allows one to write equivalent Bethe equations in terms of eigenvalues of the conserved charges, equations which have the great advantage of simply being quadratic in these new variables. These eigenvalue-based variables also allow, in section 6, to define the simple transformation which, for any given eigenstate, links its rotated basis representation to this newly proposed common framework representation and gives us, in section 7 a simple determinant representation of certain scalar products.

## 2   The XXX generalised Gaudin algebra and its algebraic Bethe Ansatz solution

The rational GGA [1, 6], which we will alternatively call XXX due to its isotropy, is an infinite dimensional Lie algebra for which the operators $S^x(u), S^y(u), S^z(u)$, parametrised by $u \in \mathbb{C}$, satisfy the following commutation relations:

$$[S^x(u), S^y(v)] = \frac{i}{u-v}(S^z(u) - S^z(v)),$$
$$[S^y(u), S^z(v)] = \frac{i}{u-v}(S^x(u) - S^x(v)),$$
$$[S^z(u), S^x(v)] = \frac{i}{u-v}(S^y(u) - S^y(v)),$$
$$[S^\kappa(u), S^\kappa(v)] = 0, \quad \kappa = x, y, z, \tag{1}$$

for any $u, v \in \mathbb{C}$ such that $u \neq v$. One can also define the operators $S^+(u) \equiv S^x(u) + iS^y(u)$ and $S^-(u) \equiv S^x(u) - iS^y(u)$, whose commutation rules are easily found to be given by:

$$\left[S^+(u), S^-(v)\right] = \frac{2\left(S^z(u) - S^z(v)\right)}{u-v},$$
$$\left[S^z(u), S^\pm(v)\right] = \pm \frac{\left(S^\pm(u) - S^\pm(v)\right)}{u-v}. \tag{2}$$

The usual QISM [5, 6] in these systems starts by defining the "transfer matrix":

$$S^2(u) = S^z(u)S^z(u) + S^x(u)S^x(u) + S^y(u)S^y(u)$$
$$= S^z(u)S^z(u) + \frac{1}{2}\left[S^+(u)S^-(u) + S^-(u)S^+(u)\right], \tag{3}$$

for which one can easily show that

$$[S^2(u), S^2(v)] = 0 \quad \forall\, u, v \in \mathbb{C}. \tag{4}$$

These operators therefore define (for arbitrary spectral parameters $u$) a family of commuting operators which share the same eigenstates. These eigenstates can be found by computing the action of the $S^2(u)$ operator on a ansatz state built as:

$$|\{\lambda_1 \ldots \lambda_M\}\rangle = \left(\prod_{m=1}^{M} S^+(\lambda_m)\right)|\Omega\rangle, \tag{5}$$

where the $S^+(\lambda_m)$ operators create excitations, parametrised by a single Bethe root $\lambda_m \in \mathbb{C}$, above a reference state $|\Omega\rangle$.

Usually, the reference state needs to be chosen to make it an eigenstate of every $S^z(u)$, i.e. $S^z(u)|\Omega\rangle = \ell^z(u)|\Omega\rangle$ as well as an eigenstate of the transfer matrix itself, i.e. $S^2(u)|\Omega\rangle = \ell(u)|\Omega\rangle$ (a condition which is equivalent to making it a highest-weight state: $S^-(u)|\Omega\rangle = 0$). If this last condition is met, the action of $S^2(u)$ on the ansatz state becomes:

$$S^2(u)|\{\lambda_1 \ldots \lambda_M\}\rangle = \left(\prod_{m=1}^{M} S^+(\lambda_m)\right)S^2(u)|\Omega\rangle + \left[S^2(u), \left(\prod_{m=1}^{M} S^+(\lambda_m)\right)\right]|\Omega\rangle$$
$$= \ell(u)|\{\lambda_1 \ldots \lambda_M\}\rangle + \sum_{p=1}^{M}\left(\prod_{m=1}^{p-1} S^+(\lambda_m)\right)\left[S^2(u), S^+(\lambda_p)\right]\left(\prod_{m=p+1}^{M} S^+(\lambda_m)\right)|\Omega\rangle, \tag{6}$$

since the pseudovaccum is an eigenstate of $S^2(u)$. From eq. (2), we know that the inserted commutators are given by:

$$
\begin{aligned}
\left[S^2(u), S^+(v)\right] &= \frac{S^+(u)S^z(v) + S^z(v)S^+(u) - S^+(v)S^z(u) - S^z(u)S^+(v)}{u-v} \\
&= 2\frac{S^+(u)S^z(v) - S^+(v)S^z(u)}{u-v}.
\end{aligned}
\tag{7}
$$

Since $\left[S^z(u), S^+(v)\right] = \frac{S^+(u) - S^+(v)}{u-v}$, the $S^z(w)$ operators which arise in the inserted commutators can all be commuted to the far right, where their action on the pseudovacuum: $S^z(w)|\Omega\rangle$ is trivially given by $\ell^z(w)|\Omega\rangle$. The importance of having an appropriately chosen pseudovacuum (eigenstate of both $S^2(u)$ and $S^z(u)$) is twofold: being an eigenstate of $S^2(u)$ allows the first term in eq. (6) to be proportional to the original ansatz state while (in conjunction with the commutation rules of the GGA) being an eigenstate of $S^z(u)$ allows us to simply rewrite the second term as a sum of similar states for which one of the $\lambda$ Bethe roots has been replaced by $u$. With these two conditions met, we find:

$$
\begin{aligned}
S^2(u)|\{\lambda_1 \ldots \lambda_M\}\rangle &= \left(\ell(u) + \sum_{p=1}^{M}\left[\frac{-2\ell^z(u)}{u-\lambda_p} + \sum_{q\neq p}^{M}\frac{1}{u-\lambda_p}\frac{1}{u-\lambda_q}\right]\right)|\{\lambda_1 \ldots \lambda_M\}\rangle \\
&+ \sum_{p=1}^{M}\frac{2}{u-\lambda_p}\left[\ell^z(\lambda_p) + \sum_{q\neq p}\frac{1}{\lambda_q - \lambda_p}\right]|\{\lambda_1 \ldots \lambda_p \to u \ldots \lambda_M\}\rangle,
\end{aligned}
\tag{8}
$$

where the state $|\{\lambda_1 \ldots \lambda_p \to u \ldots \lambda_M\}\rangle$ is built as (5), but with $\lambda_p$ replaced by the spectral parameter $u$.

The ansatz state therefore becomes one of the desired eigenstates of $S^2(u)$, with eigenvalue

$$
E(u, \lambda_1 \ldots \lambda_M) = \ell(u) + \sum_{p=1}^{M}\left[\frac{-2\ell^z(u)}{u-\lambda_p} + \sum_{q\neq p}^{M}\frac{1}{u-\lambda_p}\frac{1}{u-\lambda_q}\right],
\tag{9}
$$

whenever the unwanted terms cancel, i.e. when the Bethe roots $\lambda$ are solution to the set of $M$ Bethe equations:

$$
\ell^z(\lambda_p) + \sum_{q\neq p}^{M}\frac{1}{\lambda_q - \lambda_p} = 0.
\tag{10}
$$

For the realisations of the GGA which will be of interest in this work we will systematically have

$$
S^2(u) = \sum_{i=1}^{N}\frac{R_i}{u-\epsilon_i} + \gamma(u),
\tag{11}
$$

where $\gamma(u)$ has no singularities in $u$. Therefore, one can extract, from the $N$ poles of $S^2(u)$, a set of $N$ operator-valued residues $R_i$ which all commute with one another (since $\left[S^2(u), S^2(v)\right] = 0$) and therefore share the common eigenbasis found by the QISM. Any linear (or non-linear) combination of these conserved charges $H = \sum_{i=1}^{N}\alpha_i R_i$ is therefore an Hamiltonian whose eigenstates are those found through this procedure, and whose eigenenergies would be given by $\sum_{i=1}^{N}\alpha_i r_i$, with $r_i$ the scalar-valued residue of $E(u, \lambda_1 \ldots \lambda_M)$ at $u = \epsilon_i$, i.e. the eigenvalue of the individual conserved charge $R_i$.

## 3 Realisations of the generalised Gaudin algebra

### 3.1 Gaudin magnets in a z-oriented external magnetic field

One can build a first realisation of the GGA, using $N$-local spins defined by the usual $SU(2)$ generators $S_i^+, S_i^-, S_i^z$, by defining:

$$\tilde{S}^+(u) = \sum_{i=1}^{N} \frac{S_i^+}{u - \epsilon_i},$$

$$\tilde{S}^-(u) = \sum_{i=1}^{N} \frac{S_i^-}{u - \epsilon_i},$$

$$\tilde{S}^z(u) = -B_z - \sum_{i=1}^{N} \frac{S_i^z}{u - \epsilon_i}. \tag{12}$$

valid, in principle, for any arbitrary but distinct values of $\epsilon_i \in \mathbb{C}$ and for arbitrary $B_z \in \mathbb{C}$, although it is common to restrict both $\epsilon_i$ and $B_z$ to be real in order for the resulting conserved charges to be hermitian and lead to physical Hamiltonians. It is a simple exercise to prove, that these operators will obey the commutation rules defining the GGA (2), so that the QISM described in the previous section is directly applicable.

This first realisation defines, through the residues of $\tilde{S}^2(u)$ at $u = \epsilon_i$ (divided by the constant 2), a set of $N$ commuting conserved charges given by:

$$R_i = B_z S_i^z + \sum_{j \neq i} \frac{\vec{S}_i \cdot \vec{S}_j}{\epsilon_i - \epsilon_j}. \tag{13}$$

Taken individually, each of these conserved charges, is a central spin hamiltonian ($H_{CS} = R_1$ for example) for which the central spin (labelled 1 in this case) is coupled to and external z-oriented magnetic field $B_z$ as well as to each other spin, through an isotropic hyperfine coupling of magnitude $A_j \equiv \frac{1}{\epsilon_1 - \epsilon_j}$. Each of these coupling constants, as well as the magnetic field can be chosen as one sees fits, without affecting the QISM.

Indeed, for any given set of $\{\epsilon_1 \ldots \epsilon_N\}$ and any value of $B_z$, one can use a common pseudo vacuum: $|\Omega\rangle \equiv |\downarrow_1 \ldots \downarrow_N\rangle$, which, for spins-1/2, is the tensor product of the eigenstates of $S_i^z$ with $-1/2$ eigenvalue. For higher spins, this reference state would be the tensor product of the local $m = -J$ eigenstates with the lowest possible eigenvalue of the z-projection of the spin magnetic moment. As required, this specific state is easily shown to be an eigenstate of both $\tilde{S}^2(u)$ and $\tilde{S}^z(u)$.

Dealing, from now on, only with spins 1/2, the vacuum eigenvalues of $\tilde{S}^z(u)$ and $\tilde{S}^2(u)$ are respectively given by:

$$\ell^z(u) = -B_z + \frac{1}{2} \sum_{i=1}^{N} \frac{1}{u - \epsilon_i},$$

$$\ell(u) = \left( -B_z + \frac{1}{2} \sum_{i=1}^{N} \frac{1}{u - \epsilon_i} \right)^2 + \frac{1}{2} \sum_{i=1}^{N} \frac{1}{(u - \epsilon_i)^2}$$

$$= B_z^2 - B_z \sum_{i=1}^{N} \frac{1}{u - \epsilon_i} + \frac{1}{4} \sum_{i=1}^{N} \sum_{j \neq i}^{N} \frac{1}{(u - \epsilon_i)(u - \epsilon_j)} + \frac{3}{4} \sum_{i=1}^{N} \frac{1}{(u - \epsilon_i)^2}. \tag{14}$$

Therefore, we know that states of the form:

$$|\{\lambda_1 \ldots \lambda_M\}\rangle = \left( \prod_{m=1}^{M} \tilde{S}^+(\lambda_m) \right) |\Omega\rangle, \tag{15}$$

will be eigenstates of $S^2(u)$ with eigenvalue:

$$E(u, \lambda_1 \dots \lambda_M) = B_z^2 - B_z \sum_{i=1}^{N} \frac{1}{u - \epsilon_i} + \frac{1}{4} \sum_{i=1}^{N} \sum_{j \neq i}^{N} \frac{1}{(u - \epsilon_i)(u - \epsilon_j)} + \frac{3}{4} \sum_{i=1}^{N} \frac{1}{(u - \epsilon_i)^2}$$
$$+ \sum_{p=1}^{M} \frac{1}{u - \lambda_p} \left[ 2B_z - \sum_{i=1}^{N} \frac{1}{u - \epsilon_i} + \sum_{q \neq p}^{M} \frac{1}{u - \lambda_q} \right], \quad (16)$$

whenever the Bethe roots $\lambda$ are solution to the set of $M$ Bethe equations:

$$-B_z + \frac{1}{2} \sum_{i=1}^{N} \frac{1}{\lambda_p - \epsilon_i} + \sum_{q \neq p}^{M} \frac{1}{\lambda_q - \lambda_p} = 0. \quad (17)$$

One can finally notice that for a given set of $\{\epsilon_1 \dots \epsilon_N\}$, the resulting eigenstates are always built, independently of the value of $B_z$, using the exact same quasiparticle creation operators $\tilde{S}^+(\lambda)$ (eq. (12)) acting on the same vacuum state. This fact has lead to remarkable possibilities for computing scalar products between eigenstates of distinct Hamiltonians (differing only by the value of the magnetic field), central to the treatment of quantum quenches and other unitary time-evolution problems in these systems [7–11].

### 3.1.1 Rotated basis

Evidently, since the coupling terms in these XXX models are fully invariant under rotation, building a proper ABA for any particular orientation of the magnetic field $\vec{B} = B_x \vec{u}_x + B_y \vec{u}_y + B_z \vec{u}_z$ is in fact a simple exercise. A rotation of the chosen coordinates such that the quantisation axis used for the spin description corresponds to the direction of this field will lead to an absolutely identical problem to the one where $\vec{B} = B_z \vec{u}_z$.

In doing so, the appropriate pseudovacuum becomes, for spins $1/2$, the tensor product of the eigenstates of $\vec{B} \cdot \vec{S}_k$ with eigenvalue $-1/2$. Consequently, in the "canonical" basis of the $S_i^z$ eigenstates, the resulting Bethe eigenstates become complicated superpositions of every possible z-axis magnetisation sector: it contains the fully down-polarised, the up-polarized and states containing any number of possible spin-flips (ranging from 0 to $N$ for systems of $N$ spins $1/2$). Moreover, the relevant rotated Gaudin creation operators $S^+(u)$ become, after rotation, a linear combination of the three local operators $S_i^+, S_i^-, S_i^z$ as defined in this canonical basis. While the rotation matrix for spins is well known, the remarkable simplicity of the Bethe eigenstates is only retained, in a obvious way, within the properly oriented basis; a distinct one for each field orientation.

## 3.2 Generalisation for Gaudin magnets in a generic external magnetic field

In this work, we therefore look for a framework which allows one to write eigenstates, for arbitrarily oriented fields, in a common fashion. This is likely to facilitate the calculation, for eigenstates belonging to two distinct orientations of $\vec{B}$, of the scalar products and form factors which are at the very core of quenched dynamics. One such specific problem, for which we will derive useful simple results, is the quench from a given initial canonical basis state

$$|\uparrow_{i_1} \dots \uparrow_{i_M}\rangle \equiv \left( \prod_{j=1}^{m} S_{i_j}^+ \right) |\downarrow_1 \dots \downarrow_N\rangle, \quad (18)$$

i.e. a tensor product of local $S_i^z$ eigenstates, whose subsequent dynamics is to be driven by a central spin hamiltonian in an arbitrarily oriented finite magnetic field (or any other integrable model which shares the same eigenstates). This unitary dynamics problem requires the



capacity to project this initial state onto the eigenstates of the $\vec{B}$ integrable quantum hamiltonian. These scalar products will, within the proposed framework, have a simple representation (derived in section 7.2) which could provide major simplifications for numerical work.

The basis of this work is to include the additional $x$ and $y$ component of the magnetic field by simply adding constants to the usual realisation of the rational GGA (12), therefore defining the new realisation for $B_0^+, B_0^-$ arbitrary complex constants:

$$S^+(u) = B_0^+ + \sum_{i=1}^{N} \frac{S_i^+}{u - \epsilon_i},$$

$$S^-(u) = B_0^- + \sum_{i=1}^{N} \frac{S_i^-}{u - \epsilon_i},$$

$$S^z(u) = -B_z - \sum_{i=1}^{N} \frac{S_i^z}{u - \epsilon_i}, \tag{19}$$

expressed in terms of the unrotated local spin operators as they are usually defined using the z-quantisation axis. The presence of additional constants has, evidently, no impact on the commutation rules between these newly defined operators which therefore still trivially obey the commutation rules defining the GGA (2). Being a proper realisation of the algebra, one can define a transfer matrix:

$$S^2(u) = S^z(u)S^z(u) + \frac{1}{2}S^+(u)S^-(u) + \frac{1}{2}S^-(u)S^+(u), \tag{20}$$

which is such that $\left[S^2(u), S^2(v)\right] = 0$. Its poles, again, define a set of commuting operators which share a common eigenbasis: the $N$ underlying conserved charges found as the operator-valued residues at each $u = \epsilon_k$. The only difference in the resulting $S^2(u)$, compared the usual one (with $B_0^\pm = 0$), is the presence of the additional terms $B_0^+ \sum_{i=1}^{N} \frac{S_i^-}{u - \epsilon_i} + B_0^- \sum_{i=1}^{N} \frac{S_i^+}{u - \epsilon_i}$. The resulting conserved charges: $\frac{1}{2}$ of the residues at $u = \epsilon_i$, are then given by eq. (13) supplemented by the residue of these additional terms, namely:

$$R_i = B_z S_i^z + \frac{B_0^+}{2}S_i^- + \frac{B_0^-}{2}S_i^+ + \sum_{j \neq i} \frac{\vec{S}_i \cdot \vec{S}_j}{\epsilon_i - \epsilon_j}, \tag{21}$$

which, for the specific choice $B_0^+ = B_x + iB_y$, $B_0^- = B_x - iB_y$, become:

$$R_i = \vec{B} \cdot \vec{S}_i + \sum_{j \neq i} \frac{\vec{S}_i \cdot \vec{S}_j}{\epsilon_i - \epsilon_j}, \tag{22}$$

with a generic magnetic field $\vec{B} = B_x \vec{u}_x + B_y \vec{u}_y + B_z \vec{u}_z$. While this demonstrates the integrability of those arbitrary field Gaudin models by expliciting the existence of the $N$ commuting conserved charges, let us remind the reader that this was already obvious from the known integrability of the model with a z-oriented field and the isotropy of the coupling terms. Moreover, we already know, through this rotation, a way in which one could implement a traditional ABA for these models.

However, our aim is to now demonstrate that one can use the proposed representation (19) to build the Bethe ansatz without having to use a specific rotated basis for each possible orientation of the magnetic field.

While this specific choice of $B_0^+ = (B_0^-)^*$ leads to hermitian conserved charges, the results of sections 4 to 6 would remain valid for arbitrary non-zero values of these two parameters, by simply keeping $B_0^+, B_0^-$ ant the explicit product $B_0^+ B_0^-$ instead of replacing them with $B_x + iB_y$, $B_x - iB_y$ and $B_x^2 + B_y^2 = |B_\perp|^2$ respectively.

# 4 Common framework for Gaudin magnets in arbitrary fields

Considering the representation chosen in (19), it would impossible to properly define a reference state which is an eigenstate of both $S^2(u)$ and $S^z(u)$. Indeed, even with a single spin involved ($N = 1$), the known non-degenerate eigenstates of $S^z(u)$, namely the $|\uparrow\rangle$ and $|\downarrow\rangle$), are easily shown to not be eigenstates of $S^2(u)$ in the presence of the in-plane field terms $B_0^\pm$. While it allows for a straightforward inclusion of additional in-plane magnetic field components, the chosen representation does not provide us with a proper highest weight state to be used as a pseudovacuum. It makes it necessary to adapt the usual ABA but, as we now demonstrate, it is still actually possible to write the eigenstates for an arbitrarily oriented magnetic field as excitations over the common reference state $|\Omega\rangle \equiv |\downarrow \ldots \downarrow\rangle$:

$$|\{\lambda_1 \ldots \lambda_N\}\rangle \equiv \prod_{i=1}^{N} S^+(\lambda_i) |\Omega\rangle \tag{23}$$

using the $S^+(u)$ operator defined in eq. (19) as

$$S^+(\lambda_i) = B_x + iB_y + \sum_{i=1}^{N} \frac{S_i^+}{u - \epsilon_i}. \tag{24}$$

In the current section, we prove that these Bethe states become the common eigenstates of the set of $N$ commuting conserved charges:

$$R_k = \vec{B} \cdot \vec{S}_k + \sum_{j \neq k}^{N} \frac{\vec{S}_k \cdot \vec{S}_j}{\epsilon_k - \epsilon_j}, \tag{25}$$

(and naturally of the resulting transfer matrix $S^2(u) = \sum_k \frac{R_k}{u - \epsilon_k} + \gamma(u)$) for any set of $N$ Bethe roots which are solution to the $N$ Bethe equations:

$$-B_z + \frac{1}{2} \sum_{i=1}^{N} \frac{1}{\lambda_p - \epsilon_i} + \sum_{q \neq p}^{N} \frac{1}{\lambda_q - \lambda_p} = -\frac{(B_x^2 + B_y^2)}{2} \frac{\prod_{k=1}^{N}(\epsilon_k - \lambda_p)}{\prod_{q \neq p}^{N}(\lambda_q - \lambda_p)}, \tag{26}$$

where the corresponding eigenvalues of each $R_k$ are given, for any of these eigenstates, by:

$$r_k = -\frac{B_z}{2} + \frac{1}{4} \sum_{j \neq k}^{N} \frac{1}{\epsilon_k - \epsilon_j} - \frac{1}{2} \sum_{p=1}^{N} \frac{1}{\epsilon_k - \lambda_p}. \tag{27}$$

For real $B_x, B_y, B_z$ and real $\epsilon_i$'s, the hermiticity of the conserved charges, implies that their eigenvalues are real. Consequently, any eigenstate will always be defined in terms of Bethe roots $\lambda_i$ which are either real or come in complex conjugate pairs.

## 4.1 Quantum Inverse Scattering Method

While the "faulty" pseudovacuum $|\Omega\rangle$ remains an eigenstate of the $S^z(u)$ operator defined in (19), it no longer is an eigenstate of the resulting $S^2(u)$. This lack of a proper vacuum reference state is also typical of integrable models without U(1) symmetry and consequently a wide variety of approaches have been built in order to address this specific issue: from the diagonalisation of the XXZ open spin chain from the representation theory of the q-Onsager algebra [12], to the generalisation of the coordinate Bethe ansatz used in the XXZ chain and the antisymmetric simple exclusion process (ASEP) models in [13] as well as the functional Bethe ansatz used to treat various boundaries problems in XXX and XXZ chains [14–16].

Another systematic method recently developed to deal with the lack of proper vaccum state is the off-diagonal Bethe ansatz [17–20] which has also allowed an explicit construction of the eigenstates [21, 22] but as excitations over a more complex reference state than the simple fully polarised state used here. This is also reminiscent of the ABA treatment of the XXZ Gaudin models with open "boundaries" presented in [23], where it was shown that one can actually build a proper, though not as simple, pseudo-vacuum which makes a typical ABA possible although the resulting eigenstates are constructed in a more intricate way than in this work.

Let us mention finally that the quantum Separation of Variables (SoV) approach, originating in the work of Sklyanin [24–26], does allow for a systematic way to deal with certain integrable models when a proper ABA pseudovacuum state is not readily available [27–36]. As in this work, the eigenstates in SoV are built on the ferromagnetic reference state $|\Omega\rangle$ even when it is not appropriate as an ABA pseudovacuum (see [28] for example). However, the approach presented here stays, in spirit, as close as possible to the usual implementation of the ABA. Whereas in SoV, the eigenstates would typically be found as separate states written in an SoV basis [28], in this work they will be explicitly built, just like in the ABA, in the very compact form of the repeated action of quasi-particle creation operators $S^+(\lambda)$.

This work follows the approach originating in [37], where it was shown how, despite a similarly "faulty" pseudovacuum, that Bethe equations, eigenvalues and eigenstates can be found for the dual representation of integrable spin-boson Gaudin-like models. Later, the same idea was also used to explicitly construct the eigenstates of $p+ip$ pairing Hamiltonian whose interaction with an environnement breaks the U(1) symmetry [38], a work which complements [39], where the conserved charges, Bethe equations and the energy spectrum were found through the boundary quantum inverse scattering method.

First, let us again notice that the eigenstates, as we know from the rotated framework, will have contributions coming from every possible value of the total z-axis magnetisation. Consequently, an ansatz state of the form $\prod_{i=1}^{M} S^+(\lambda_i)|\Omega\rangle$ can only be one of the eigenstates if it systematically contains $M = N$ excitations, hence the proposed $N$ Bethe roots ansatz state found in (23), a fact which was also observed in XXX spin chains with general boundaries [40].

The action of the transfer matrix on this ansatz state is, again, given by

$$S^2(u)\left(\prod_{i=1}^{N} S^+(\lambda_i)\right)|\Omega\rangle = \left(\prod_{i=1}^{N} S^+(\lambda_i)\right)S^2(u)|\Omega\rangle + \left[S^2(u), \prod_{i=1}^{N} S^+(\lambda_i)\right]|\Omega\rangle, \qquad (28)$$

just as was the case before in eq. (6).

The evaluation of the second term $\left[S^2(u), \prod_{i=1}^{N} S^+(\lambda_i)\right]|\Omega\rangle$, relies exclusively on the commutation rules of the GGA and on the fact that the pseudovacuum is an eigenstate of $S^z(u)$. Both of these conditions are, once again, met here and consequently the calculation proceeds exactly as in section 2 giving us again:

$$\left[S^2(u), \prod_{i=1}^{N} S^+(\lambda_i)\right]|\Omega\rangle = \left(\sum_{p=1}^{N}\left[\frac{-2\ell^z(u)}{u-\lambda_p} + \sum_{q\neq p}^{N}\frac{1}{u-\lambda_p}\frac{1}{u-\lambda_q}\right]\right)|\{\lambda_1\ldots\lambda_N\}\rangle$$
$$+ \sum_{p=1}^{N}\frac{2}{u-\lambda_p}\left[\ell^z(\lambda_p) + \sum_{q\neq p}^{N}\frac{1}{\lambda_q-\lambda_p}\right]|\{\lambda_1\ldots\lambda_p \to u\ldots\lambda_N\}\rangle, \quad (29)$$

The first term in (28) however appears to be problematic since $|\Omega\rangle$ is no longer an eigenstate of $S^2(u)$. In fact, a simple calculation gives:

$$S^2(u)|\Omega\rangle = \left(\ell(u) + B_0^+ B_0^-\right)|\Omega\rangle + B_0^- S^+(u)|\Omega\rangle = \left(\ell(u) + |B_\perp|^2\right)|\Omega\rangle + B_0^- S^+(u)|\Omega\rangle. \qquad (30)$$

All in all, one finds the action of the resulting transfer matrix to be given by:

$$S^2(u)\left(\prod_{i=1}^N S^+(\lambda_i)\right)|\Omega\rangle = E(u,\{\lambda_1\ldots\lambda_N\})\left(\prod_{i=1}^N S^+(\lambda_i)\right)|\Omega\rangle + B_0^- S^+(u)\left(\prod_{i=1}^N S^+(\lambda_i)\right)|\Omega\rangle$$

$$+ \sum_{p=1}^N F_p(u,\{\lambda_1\ldots\lambda_N\})\left(\prod_{i\neq p}^N S^+(\lambda_i)\right)S^+(u)|\Omega\rangle, \quad (31)$$

with

$$E(u,\{\lambda_1\ldots\lambda_N\}) = \left(\ell(u) + |B_\perp|^2 + \sum_{p=1}^N\left[\frac{-2\ell^z(u)}{u-\lambda_p} + \sum_{q\neq p}^N \frac{1}{u-\lambda_p}\frac{1}{u-\lambda_q}\right]\right),$$

$$F_p(u,\{\lambda_1\ldots\lambda_N\}) = \frac{2}{u-\lambda_p}\left[\ell^z(\lambda_p) + \sum_{q\neq p}^N \frac{1}{\lambda_q-\lambda_p}\right], \quad (32)$$

which is mostly similar, in form, to eq. (8) except that now $M = N$ and the additional constant $|B_\perp|^2$ has been added to $\ell(u)$ (the vacuum eigenvalue of the original z-field transfer matrix). The only major difference is the presence of the additional term $B_0^- S^+(u)\left(\prod_{i=1}^N S^+(\lambda_i)\right)|\Omega\rangle$, due to which, it is no longer possible to simply read from eq. (31) what the Bethe equations should be in order to make the proposed ansatz an eigenstate of the transfer matrix.

The specific case with $B_0^- = 0$ are worth mentioning explicitly now, since they would actually correspond to the case where the fully down-polarised vacuum is a proper ABA pseudovaccum. In this particular case, independently of the value of $B_0^+$, the resulting action of the transfer matrix leads to the exact same form as in the z-oriented field case discussed in section 3.1. Therefore, the proper ansatz states would have to be defined with a fixed number $M \leq N$ of Bethe roots $\prod_{i=1}^M S^+(\lambda_i)|\Omega\rangle$ and solutions to the Bethe equation are to be found in every possible sector $M \in [0,N]$. For $B_0^+ = 0$ (the usual z-field case) the eigenstates have a fixed magnetisation sector, i.e. $M$ spins pointing up but, for $B_0^+ \neq 0$, the $S^+(u)$ operators would now contain a non-zero constant term $B_0^+$ so that the eigenstates become a superposition of the magnetisation sector containing $0, 1, 2\ldots M$ up-pointing spins, limited to a maximal magnetisation given by $M$ up-spins. This type of construction is in direct correspondance with the quasi-classical limit of the XXX-spin chain with triangular boundary conditions treated in [41], where a similar observation was made.

In the generic $B_0^+, B_0^- \neq 0$ case, dealing with the additional term in (31) is the main issue of concern, one which will be addressed in the next section by looking at the action of individual conserved charges instead of the full transfer matrix. In other integrable systems, the same issue has been dealt with by writing the explicit action of the additional operator in a way which brings (31) back to the form (8) but with new eigenvalue and unwanted coefficients whose cancelling lead to a new set of inhomogeneous Bethe equations. This type of Modified Algebraic Bethe Ansatz approach has, amongst others been carried out for the totally antisymmetric simple exclusion process (TASEP) [42], for XXZ chains with boundaries [43] and for twisted XXX chains [44].

## 4.2 Building the Bethe equations

As demonstrated in [37, 38], one way to circumvent this "faulty vacuum" difficulty is to look, not at the transfer matrix $S^2(u)$ as a whole, but at the action of individual conserved charges $R_k$ on the ansatz state. These are found by simply looking at $\frac{1}{2}$ of the $u = \epsilon_k$ residues of eq.

(31), i.e.:

$$R_k \left( \prod_{i=1}^{N} S^+(\lambda_i) \right) |\Omega\rangle = \mathrm{Res} \left( \frac{E(u, \{\lambda_1 \dots \lambda_N\})}{2}, \epsilon_k \right) \left( \prod_{i=1}^{N} S^+(\lambda_i) \right) |\Omega\rangle$$

$$+ \sum_{p=1}^{N} \mathrm{Res} \left( \frac{F_p(u, \{\lambda_1 \dots \lambda_N\})}{2} S^+(u), \epsilon_k \right) \left( \prod_{i \neq p}^{N} S^+(\lambda_i) \right) |\Omega\rangle$$

$$+ B_0^- \, \mathrm{Res} \left( \frac{S^+(u)}{2}, \epsilon_k \right) \left( \prod_{i=1}^{N} S^+(\lambda_i) \right) |\Omega\rangle . \quad (33)$$

All of the residues are easily evaluated:

$$\mathrm{Res}(S^+(u), \epsilon_k)) = S_k^+,$$

$$\mathrm{Res}\left( F_p \left( u, \{\lambda_1 \dots \lambda_N\} S^+(u) \right), \epsilon_k \right) = \left[ -B_z + \frac{1}{2} \sum_{i=1}^{N} \frac{1}{\lambda_p - \epsilon_i} + \sum_{q \neq p}^{N} \frac{1}{\lambda_q - \lambda_p} \right] \frac{2 S_k^+}{\epsilon_k - \lambda_p},$$

$$\mathrm{Res}\left( E\left( u, \{\lambda_1 \dots \lambda_N\} \right), \epsilon_k \right) = -B_z + \frac{1}{2} \sum_{j \neq k}^{N} \frac{1}{\epsilon_k - \epsilon_j} - \sum_{p=1}^{N} \frac{1}{\epsilon_k - \lambda_p}, \quad (34)$$

since we know, from eq. (14), that

$$\mathrm{Res}(\ell^z(u), \epsilon_k) = \frac{1}{2},$$

$$\mathrm{Res}(\ell(u), \epsilon_k) = -B_z + \frac{1}{2} \sum_{j \neq k}^{N} \frac{1}{\epsilon_k - \epsilon_j}. \quad (35)$$

This results in

$$R_k \left( \prod_{i=1}^{N} S^+(\lambda_i) \right) |\Omega\rangle = r_k \left( \prod_{i=1}^{N} S^+(\lambda_i) \right) |\Omega\rangle + \frac{B_0^-}{2} S_k^+ \left( \prod_{i=1}^{N} S^+(\lambda_i) \right) |\Omega\rangle$$

$$+ \sum_{p=1}^{N} \left[ -B_z + \frac{1}{2} \sum_{i=1}^{N} \frac{1}{\lambda_p - \epsilon_i} + \sum_{q \neq p}^{N} \frac{1}{\lambda_q - \lambda_p} \right] \frac{S_k^+}{\epsilon_k - \lambda_p} \left( \prod_{i \neq p}^{N} S^+(\lambda_i) \right) |\Omega\rangle , \quad (36)$$

with

$$r_k = -\frac{B_z}{2} + \frac{1}{4} \sum_{j \neq k}^{N} \frac{1}{\epsilon_k - \epsilon_j} - \frac{1}{2} \sum_{p=1}^{N} \frac{1}{\epsilon_k - \lambda_p}. \quad (37)$$

Defining

$$\Gamma_p \equiv -B_z + \frac{1}{2} \sum_{i=1}^{N} \frac{1}{\lambda_p - \epsilon_i} + \sum_{q \neq p} \frac{1}{\lambda_q - \lambda_p}, \quad (38)$$

the form given by eq (36) allows us to build proper Bethe equations by finding sets of Bethe roots which cancel the last two terms:

$$\frac{B_0^-}{2} \left( \prod_{i=1}^{N} S^+(\lambda_i) \right) S_k^+ |\Omega\rangle + \sum_{p=1}^{N} \frac{\Gamma_p}{\epsilon_k - \lambda_p} \left( \prod_{i \neq p}^{N} S^+(\lambda_i) \right) S_k^+ |\Omega\rangle = 0, \quad (39)$$

for each and every $k = 1, \ldots N$. Doing so would indeed build an eigenstate common to all the conserved charges $R_k$ (with eigenvalue $r_k$) which is precisely what we want to do. These unwanted terms (39) can be decomposed on the set of basis states $\left(\prod_{k=1}^M S_{i_k}^+\right) S_k^+ |\Omega\rangle$, for which spin $k$ is always pointing up while an arbitrary subset of $M$ other spins are also pointing up. Due to the constant term $B_0^+$ in $S^+(\lambda)$, any number of additional flipped spins $0 \le M \le N-1$ is possible. In order to find the appropriate Bethe equations, we first find the set of equations which would cancel the coefficient in front of the basis state $S_k^+ |\Omega\rangle$, stemming from $M = 0$ additional spin flips, i.e. using the constant part $B_0^+$ of every $S^+(\lambda_i)$ operator in eq. (39). Namely, we require

$$\frac{B_0^-}{2}\left(B_0^+\right)^N + \left(B_0^+\right)^{N-1} \sum_{p=1}^N \frac{\Gamma_p}{\epsilon_k - \lambda_p} = 0 \;\rightarrow\; \sum_{p=1}^N \frac{\Gamma_p}{\epsilon_k - \lambda_p} = -\frac{|B_\perp|^2}{2}. \tag{40}$$

Cancelling simultaneously these $N$ terms (one for each $k = 1, \ldots N$), leads us to a Cauchy matrix system for the $N$ values of $\Gamma_p$ (and for the $N$ Bethe roots which ultimately define them):

$$\begin{pmatrix} \frac{1}{\epsilon_1-\lambda_1} & \frac{1}{\epsilon_1-\lambda_2} & \cdots & \frac{1}{\epsilon_1-\lambda_N} \\ \frac{1}{\epsilon_2-\lambda_1} & \frac{1}{\epsilon_2-\lambda_2} & \cdots & \frac{1}{\epsilon_2-\lambda_N} \\ \vdots & \ddots & \vdots \\ \frac{1}{\epsilon_N-\lambda_1} & \frac{1}{\epsilon_N-\lambda_2} & \cdots & \frac{1}{\epsilon_N-\lambda_N} \end{pmatrix} \begin{pmatrix} \Gamma_1 \\ \Gamma_2 \\ \vdots \\ \Gamma_N \end{pmatrix} = \begin{pmatrix} -|B_\perp|^2/2 \\ -|B_\perp|^2/2 \\ \vdots \\ -|B_\perp|^2/2 \end{pmatrix}, \tag{41}$$

whose solution, using the well-known inverse of the Cauchy matrix, is easily found to be given by:

$$\Gamma_p = -B_z + \frac{1}{2}\sum_{i=1}^N \frac{1}{\lambda_p - \epsilon_i} + \sum_{q \ne p} \frac{1}{\lambda_q - \lambda_p} = -\frac{|B_\perp|^2}{2}\frac{\prod_{k=1}^N(\epsilon_k - \lambda_p)}{\prod_{q \ne p}(\lambda_q - \lambda_p)}. \tag{42}$$

This set of $N$ equations will be proper Bethe equations, provided it is shown that sets of $\Gamma_p$ which solve them also cancel each and every other coefficient which appears in the unwanted terms (39); a fact we demonstrate in the next subsection. Let us point out that similar inhomogeneous Bethe equations, with an additional product term on the right hand side, seem characteristic of Gaudin models without U(1) symmetry having been found previously in [38, 39, 45, 46].

### 4.3 Validity of the proposed Bethe equations

In the unwanted term (39), the generic coefficient $C_{\{i_1 \ldots i_M, k\}}$ found in front of a given basis state $|\uparrow_{i_1} \ldots \uparrow_k \ldots \uparrow_{i_M}\rangle$ containing $M+1$ flipped-up spins (including spin $k$ with certainty), can be written as:

$$C_{\{i_1 \ldots i_M, k\}} = \left(B_0^+\right)^{N-M-1}\left(\frac{B_0^-}{2}B_0^+ \sum_{L \in S^M}\left(\prod_{j=1}^M \frac{1}{\ell_j - \epsilon_{i_j}}\right) + \sum_{p=1}^N \frac{\Gamma_p}{\epsilon_k - \lambda_p}\left[\sum_{L_{\hat{p}} \in S_{\hat{p}}^M}\left(\prod_{j=1}^M \frac{1}{\ell_j - \epsilon_{i_j}}\right)\right]\right), \tag{43}$$

where $L = (\ell_1, \ell_2 \ldots \ell_M)$ is an element of $S^M$: the set composed of every permutation of every subset of $\{\lambda_1 \ldots \lambda_N\}$ with cardinality $M$. On the other hand, $L_{\hat{p}} = (\ell_1, \ell_2 \ldots \ell_M)$ is an element of $S_{\hat{p}}^M$, the set which is built like $S^M$ but from permutation of M-cardinality subsets of the $N-1$ Bethe roots which excludes $\lambda_p$.

That is to say that coefficients are built by flipping each of the $M-1$ spins $i_k$ using the $\frac{S_{i_k}^+}{\lambda_i - \epsilon_{i_k}}$ part of one specific associated $S^+(\lambda_i)$. The remaining $S^+(\lambda_i)$, not used to flip spins,

then contribute $B_0^-$ coming from their constant term. We finally need to sum the possible ways to make this bijective association between the $M$ up spins and the associated $M$ Bethe roots used to flip them.

The second term in (43) can be rewritten as a full sum over $S^M$ (which then includes $\lambda_p$) from which we remove the set of terms which do necessarily contain $\lambda_p$, i.e.: the elements of $S^M \setminus S_{\hat{p}}^M$. Doing so gives:

$$
\frac{C_{\{i_1...i_M,k\}}}{\left(B_0^+\right)^{N-M-1}} = \left( \frac{|B_\perp|^2}{2} \sum_{L\in S^M} \left( \prod_{j=1}^M \frac{1}{\ell_j - \epsilon_{i_j}} \right) \right.
$$
$$
\left. + \sum_{p=1}^N \frac{\Gamma_p}{\epsilon_k - \lambda_p} \left[ \sum_{L\in S^M} \left( \prod_{j=1}^M \frac{1}{\ell_j - \epsilon_{i_j}} \right) - \sum_{L\in S^M \setminus S_{\hat{p}}^M} \left( \prod_{j=1}^M \frac{1}{\ell_j - \epsilon_{i_j}} \right) \right] \right). \quad (44)
$$

For Bethe roots solution to the proposed Bethe equations (40), the first two sums then cancel out, since $\sum_{p=1}^N \frac{\Gamma_p}{\epsilon_k - \lambda_p} = -\frac{|B_\perp|^2}{2}$, and the equation reduces to:

$$
\frac{C_{\{i_1...i_M,k\}}}{\left(B_0^+\right)^{N-M-1}} = -\sum_{p=1}^N \frac{\Gamma_p}{\epsilon_k - \lambda_p} \left[ \sum_{L\in S^M \setminus S_{\hat{p}}^M} \left( \prod_{j=1}^M \frac{1}{\ell_j - \epsilon_{i_j}} \right) \right], \quad (45)
$$

where each of the terms now contain, with certainty, an element $\ell_j = \lambda_p$. In each term of the sum $\lambda_p$ is paired with one of the given $\epsilon_{i_j}$. The one which is paired with $\lambda_p$ will now be called $\epsilon_{i_{k'}}$ and the sum can then be rewritten, by taking out the $\lambda_p$ term, as:

$$
\frac{C_{\{i_1...i_M,k\}}}{\left(B_0^+\right)^{N-M-1}} = -\sum_{k'=1}^M \sum_{p=1}^N \frac{\Gamma_p}{(\epsilon_k - \lambda_p)(\lambda_p - \epsilon_{i_{k'}})} \left[ \sum_{L\in S_{\hat{p}}^{M-1}} \left( \prod_{j\neq k'}^M \frac{1}{\ell_j - \epsilon_{i_j}} \right) \right]
$$
$$
= -\sum_{k'=1}^M \sum_{p=1}^N \frac{\Gamma_p}{(\epsilon_k - \lambda_p)(\lambda_p - \epsilon_{i_{k'}})} \left[ \sum_{L\in S^{M-1}} \left( \prod_{j\neq k'}^M \frac{1}{\ell_j - \epsilon_{i_j}} \right) \right.
$$
$$
\left. - \sum_{L\in S^{M-1} \setminus S_{\hat{p}}^{M-1}} \left( \prod_{j\neq k'}^M \frac{1}{\ell_j - \epsilon_{i_j}} \right) \right], \quad (46)
$$

where, once again, the sum was extended to $S^{M-1}$ by adding terms containing $\lambda_p$ and subtracting the elements of $S^{M-1} \setminus S_{\hat{p}}^{M-1}$ which contain $\lambda_p$ with certainty.

The first term is now proportional to $\sum_{p=1}^N \frac{\Gamma_p}{(\epsilon_k - \lambda_p)(\lambda_p - \epsilon_{k'})}$ while the second one has the same form as (45) but with $\epsilon_{i_{k'}}$ no longer present. One can keep the procedure going by writing this last term as:

$$
\sum_{k'=1}^M \sum_{p=1}^N \frac{\Gamma_p}{(\epsilon_k - \lambda_p)(\lambda_p - \epsilon_{i_{k'}})} \sum_{L\in S^{M-1} \setminus S_{\hat{p}}^{M-1}} \left( \prod_{j\neq k'}^M \frac{1}{\ell_j - \epsilon_{i_j}} \right)
$$
$$
= \sum_{k'=1}^M \sum_{k''\neq k'}^M \sum_{p=1}^N \frac{\Gamma_p}{(\epsilon_k - \lambda_p)(\lambda_p - \epsilon_{i_{k'}})(\lambda_p - \epsilon_{i_{k''}})} \left[ \sum_{L\in S^{M-2}} \left( \prod_{j\neq k',k''}^M \frac{1}{\ell_j - \epsilon_{i_j}} \right) \right.
$$
$$
\left. - \sum_{L\in S^{M-2} \setminus S_{\hat{p}}^{M-2}} \left( \prod_{j\neq k',k''}^M \frac{1}{\ell_j - \epsilon_{i_j}} \right) \right], \quad (47)
$$

with the first term now proportional to $\sum_{k'=1}^{M} \sum_{k''\neq k'}^{M} \sum_{p=1}^{N} \frac{\Gamma_p}{(\epsilon_k - \lambda_p)(\lambda_p - \epsilon_{i_{k'}})(\lambda_p - \epsilon_{i_{k''}})}$, while the second term has again the same form as (45). Repeating the process until $\lambda_p$ has been taken out of the sum $M$ times, the resulting expressions for each of the coefficients will be given by a sum of terms proportional to:

$$\sum_{p=1}^{N} \frac{\Gamma_p}{\prod_{j=1}^{r}(\lambda_p - \epsilon_{k_j})}. \tag{48}$$

Only terms with $r \geq 2$ are involved since at least two $\epsilon$'s will appear (paired with $\lambda_p$) in the denominator of the terms defining the coefficients. We now prove that all of the terms defined by (48) with $r \geq 2$ are strictly equal to 0 when $\Gamma_p$ are solution to the proposed Bethe equations (40). Doing so, proves that these solutions cancel out completely the unwanted term (39) and therefore define the proper eigenstates of the system.

For solutions to our proposed Bethe equations (42), each of the terms defined by (48) would also be given by

$$\sum_{p=1}^{N} \frac{\Gamma_p}{\prod_{j=1}^{r}(\lambda_p - \epsilon_{k_j})} = -\frac{|B_\perp|^2}{2} \sum_{p=1}^{N} \frac{\prod_{k=1}^{N}(\lambda_p - \epsilon_k)}{\prod_{j=1}^{r}(\lambda_p - \epsilon_{k_j})\prod_{q\neq p}^{N}(\lambda_q - \lambda_p)}. \tag{49}$$

A general polynomial $P(z)$ of maximal degree $N-1$ can be decomposed exactly, using the $N$ nodes $\lambda_p$, into Lagrange polynomials:

$$P(z) = \ell(z) \sum_{p=1}^{N} \frac{P(\lambda_p)}{z - \lambda_p} \frac{1}{\prod_{q\neq p}^{N}(\lambda_q - \lambda_p)}, \tag{50}$$

where $\ell(z) = \prod_{m=1}^{N}(z - \lambda_m)$. Therefore, the polynomial

$$A(z) = \frac{\prod_{k=1}^{N}(z - \epsilon_k)}{\prod_{j=1}^{r-1}(z - \epsilon_{k_j})}, \tag{51}$$

which has maximal order $N-1$ (for $r \geq 2$), can be written exactly as:

$$\frac{A(z)}{\ell(z)} = \sum_{p=1}^{N} \frac{1}{z - \lambda_p} \frac{\prod_{k=1}^{N}(\lambda_p - \epsilon_k)}{\prod_{j=1}^{r-1}(\lambda_p - \epsilon_{k_j})\prod_{q\neq p}^{N}(\lambda_q - \lambda_p)}. \tag{52}$$

Setting $z = \epsilon_{k_r}$, we immediately see that it completes the product on $j$ which now goes from 1 to $r$. Eq. (49) can therefore be written as:

$$\sum_{p=1}^{N} \frac{\Gamma_p}{\prod_{j=1}^{r}(\lambda_p - \epsilon_{k_j})} = \frac{|B_\perp|^2}{2} \frac{A(\epsilon_{k_r})}{\ell(\epsilon_{k_r})} = 0. \tag{53}$$

It is indeed equal to 0, since $A(z)$ does have a zero at $z = \epsilon_{k_r}$ which was not removed by the denominator in its definition made in eq. (51).

Consequently, for sets of Bethe roots solution to eqs (26), will cancel the unwanted terms (39), proving the statement captured in eqs (23) to (27).

# 5 Quadratic Bethe equations

Having shown that the $N$ Bethe roots define an eigenstate when they are a solution of the Bethe equations:

$$-B_z + \frac{1}{2}\sum_{i=1}^{N}\frac{1}{\lambda_p - \epsilon_i} + \sum_{q\neq p}^{N}\frac{1}{\lambda_q - \lambda_p} = -\frac{|B_\perp|^2}{2}\frac{\prod_{k=1}^{N}(\epsilon_k - \lambda_p)}{\prod_{q\neq p}^{N}(\lambda_q - \lambda_p)}, \tag{54}$$

one can now make a change of variables:

$$\Lambda_i \equiv \sum_{p=1}^{N}\frac{1}{\epsilon_i - \lambda_p}, \tag{55}$$

to find that eigenstates can also be defined by a set $\{\Lambda_1 \dots \Lambda_N\}$ which is solution to a set of $N$ quadratic Bethe equations [47]. As can be seen from (27), they correspond to the non-trivial (state dependent) part of the conserved charges eigenvalues. These ideas have already been described and used in a variety of GGA based models [37,48–51]. It provides, just like the Heine-Stieltjes polynomial approach [52–59], major numerical simplifications in the finding of eigenstates. Not only are the resulting Bethe equations simpler since they are quadratic, they are further simplified by the fact that their solutions will be restricted to $\Lambda_i \in \mathbb{R}$ which is a consequence of the previously mentioned fact that Bethe roots defining eigenstates can only be real or come in complex conjugate pairs.

In order to find the proper set of quadratic equations, we first define $\Lambda(z) \equiv \sum_{p=1}^{N}\frac{1}{z-\lambda_p}$ as the logarithmic derivative of the polynomial $Q(z) = \prod_{p=1}^{N}(z-\lambda_p)$. In general, such a rational function with $N$ simple poles (of residue 1) placed at arbitrary points $\lambda_p$ is easily shown to be such that:

$$\Lambda(z)^2 + \Lambda'(z) = \sum_{p}\sum_{q\neq p}\frac{2}{(z-\lambda_p)(\lambda_p - \lambda_q)}. \tag{56}$$

Provided the poles are now placed at a set of $\lambda_p$ which forms a solution to eq. (54), the sum over $q$ can be replaced to give:

$$\Lambda(z)^2 + \Lambda'(z) = \sum_{p=1}^{N}\frac{2}{(z-\lambda_p)}\left(\frac{|B_\perp|^2}{2}\frac{\prod_{k=1}^{N}(\epsilon_k - \lambda_p)}{\prod_{q\neq p}^{N}(\lambda_q - \lambda_p)} - B_z + \frac{1}{2}\sum_{q=1}^{N}\frac{1}{\lambda_p - \epsilon_q}\right). \tag{57}$$

One can then find the quadratic Bethe equations in $\Lambda$, by taking the limit $z \to \epsilon_i$ of this last equation ($\forall\, i = 1 \dots N$), which first gives the $N$ equations:

$$\Lambda_i^2 - \sum_{j\neq i}^{N}\frac{\Lambda_i - \Lambda_j}{\epsilon_i - \epsilon_j} + 2B_z\Lambda_i = |B_\perp|^2\left(\sum_{p=1}^{N}\frac{\prod_{k\neq i}^{N}(\epsilon_k - \lambda_p)}{\prod_{q\neq p}^{N}(\lambda_q - \lambda_p)}\right). \tag{58}$$

This can be vastly simplified by showing that, for arbitrary values of $\epsilon$ and $\lambda$, the term $\sum_{p=1}^{N}\frac{\prod_{k\neq i}^{N}(\epsilon_k - \lambda_p)}{\prod_{q\neq p}^{N}(\lambda_q - \lambda_p)} = 1$. This last affirmation is easily demonstrated by realising that:

$$A(z) - \tilde{A}(z) = Q(z), \tag{59}$$

where $A(z) = \prod_{k=1}^{N}(z-\epsilon_k)$ is the $N^{\text{th}}$ degree polynomial with its zeros placed at each $\epsilon_i$ and $\tilde{A}(z)$ is the unique polynomial of degree $N-1$ such that it has the same value as $A(z)$ at each of the $N$ zeros of $Q(z) = \prod_{p=1}^{N}(z-\lambda_p)$, i.e. $\tilde{A}(\lambda_p) = A(\lambda_p)\,\forall\, p = 1 \dots N$.

Considering that the polynomial $P(z) = A(z) - \tilde{A}(z)$ is of degree $N$ and it has its $N$ zeros at each of the $z = \lambda_p$, $P(z)$ has to be given by $C\,Q(z)$, i.e. the only polynomials of order $N$ whose $N$ zeros are at $\lambda_p$. The proportionality constant $C$ can then be fixed to $C = 1$ by looking at the coefficient in $z^N$ which, for $A(z)$ (and therefore for $P(z)$ as well) is equal to 1 just as is the case for $Q(z)$. Consequently, since $A(\epsilon_i) = 0$ we have

$$P(\epsilon_i) = -\tilde{A}(\epsilon_i) = Q(\epsilon_i). \tag{60}$$

On the other hand, the Lagrange decomposition of $\tilde{A}(z)$ on the $N$ nodes $\{\lambda_1 \ldots \lambda_N\}$ is exact since it is a polynomial of order $N - 1$ and is given, since $\tilde{A}(\lambda_p) = A(\lambda_p)$, by both:

$$\tilde{A}(z) = Q(z) \sum_{p=1}^{N} \frac{1}{z - \lambda_p} \frac{\tilde{A}(\lambda_p)}{Q'(\lambda_p)} = Q(z) \sum_{p=1}^{N} \frac{1}{z - \lambda_p} \frac{A(\lambda_p)}{Q'(\lambda_p)}. \tag{61}$$

Evaluated at $z = \epsilon_i$, this indicates that:

$$\tilde{A}(\epsilon_i) = Q(\epsilon_i) \sum_{p=1}^{N} \frac{1}{\epsilon_i - \lambda_p} \frac{\prod_{k=1}^{N}(\lambda_p - \epsilon_k)}{\prod_{q \neq p}^{N}(\lambda_p - \lambda_q)}. \tag{62}$$

Since we showed in eq. (60) that $\tilde{A}(\epsilon_i) = -Q(\epsilon_i)$, this completes the proof that

$$\sum_{p=1}^{N} \frac{1}{\lambda_p - \epsilon_i} \frac{\prod_{k=1}^{N}(\lambda_p - \epsilon_k)}{\prod_{q \neq p}^{N}(\lambda_p - \lambda_q)} = 1 \quad \forall\, i = 1 \ldots N, \tag{63}$$

is true for arbitrary sets $\{\epsilon_1 \ldots \epsilon_N\}$ and $\{\lambda_1 \ldots \lambda_N\}$. Consequently, sets of $\{\lambda_1 \ldots \lambda_N\}$ solution to the Bethe equations (54) can also be defined through the corresponding sets of real-valued $\{\Lambda_1 \ldots \Lambda_N\}$ solution to (58)), which become the following set of $N$ quadratic Bethe equations:

$$\Lambda_i^2 - \sum_{j \neq i}^{N} \frac{\Lambda_i - \Lambda_j}{\epsilon_i - \epsilon_j} + 2B_z \Lambda_i = |B_\perp|^2. \tag{64}$$

# 6 Correspondence between the rotated basis and the common basis

The rotated basis provides us with the standard implementation of the ABA presented in section 2 while the common framework leads to slightly modified Bethe equations and eigenstates defined in terms of $N$ instead of $M < N$ Bethe roots. Using the variables $\Lambda_i \equiv \sum_{p=1}^{N} \frac{1}{\epsilon_i - \lambda_p}$, one can easily find the corresponding eigenstate in the rotated basis, where it is also defined in terms of the $N$ variables $\tilde{\Lambda}_i \equiv \sum_{p=1}^{M} \frac{1}{\epsilon_i - \mu_p}$ built out of only $M$ Bethe roots $\{\mu_1 \ldots \mu_M\}$.

Indeed, any eigenstate in both representations needs to have the same eigenvalue of the $N$ conserved charges. The association of a solution $\{\lambda_1 \ldots \lambda_N\}$ in the common framework to its corresponding rotated framework solution $\{\mu_1 \ldots \mu_M\}$ is then easily achieved by enforcing the equality of the $N$ eigenvalues $r_k$.

In the rotated basis, the usual ABA applies, and the $M$ Bethe roots have to be solution of the quadratic Bethe equations [48]:

$$\tilde{\Lambda}_i^2 - \sum_{j \neq i}^{N} \frac{\tilde{\Lambda}_i - \tilde{\Lambda}_j}{\epsilon_i - \epsilon_j} + 2|B|\,\tilde{\Lambda}_i = 0, \tag{65}$$

whose solutions are such that $\tilde{\Lambda}_i \in \mathbb{R}$, while in the common framework, we have seen that the $N$ Bethe roots are solution to:

$$\Lambda_i^2 - \sum_{j \neq i}^{N} \frac{\Lambda_i - \Lambda_j}{\epsilon_i - \epsilon_j} + 2B_z \Lambda_i = |B_\perp|^2, \tag{66}$$

whose solutions are also guaranteed to lead to real-valued $\Lambda_i$. In the rotated basis we have, from $\frac{1}{2}$ the residues of the transfer matrix' eigenvalue (9):

$$r_k = -\frac{|B|}{2} + \frac{1}{4} \sum_{j \neq k}^{N} \frac{1}{\epsilon_k - \epsilon_j} - \frac{1}{2} \tilde{\Lambda}_k, \tag{67}$$

while in the common framework they are given by:

$$r_k = -\frac{B_z}{2} + \frac{1}{4} \sum_{j \neq k}^{N} \frac{1}{\epsilon_k - \epsilon_j} - \frac{1}{2} \Lambda_k. \tag{68}$$

It is then trivial to see that the eigenstate defined in the common framework by the $N$ variables $\Lambda_j = \sum_{k=1}^{N} \frac{1}{\epsilon_j - \lambda_k}$ is the same eigenstate whose rotated basis representation is given by the set $\tilde{\Lambda}_j = \sum_{k=1}^{M} \frac{1}{\epsilon_j - \mu_k}$ related by the simple transformation:

$$\tilde{\Lambda}_j = \Lambda_j + B_z - |B|. \tag{69}$$

With $\theta$ the azimutal angle the magnetic field makes with the $z$-axis, one can also write this simple shift, defining the correspondence, as:

$$\tilde{\Lambda}_j = \Lambda_j + |B| \left( \cos(\theta) - 1 \right). \tag{70}$$

From the point of view of the actual Bethe roots $\lambda$ or $\mu$, this transformation relates sets of $N$ roots $\lambda$ to sets of $M$ roots $\mu$ in a highly non-trivial way. However, using these $\Lambda$ variables (which define the state-dependent parts of the conserved charges' eigenvalues) the correspondence becomes remarkably simple. For consistency, one can also easily check that this shift does indeed transform the quadratic equations (66) into (65) and vice-versa.

There is therefore an extraordinarily simple homotopy relating the solutions in the rotated basis and the common framework, namely just a common global shift of the $\Lambda$ variables and therefore of the full eigenspectrum. Since the rotated basis quadratic equation has been demonstrated to give a complete set of eigenstates for generic values of the system parameters [60], the same holds true in the present case. Indeed, the completeness of the $\Lambda$ solutions to the original (rotated basis) set of quadratic Bethe equations [60] relies on the fact that it has a simple (non-degenerate) spectrum, a fact which was also shown in [34]. The completeness of the proposed Bethe Ansatz is therefore trivially carried over, by this simple one to one correspondence, to the solutions of (66). Since they are a simple shift of each and every eigenvalue, this modification cannot introduce any degeneracies nor extraneous solutions, making this formulation complete for arbitrary $|B_\perp|^2$, i.e. for arbitrary external fields.

Since, in this common framework, $N$ Bethe roots are systematically used to define states of the form (23), arbitrary values of the $N$ variables $\Lambda_j$ will always correspond to some set of $N$ values of $\lambda_k$. Therefore, arbitrary $\Lambda_j$ always define a generic off-the-shell Bethe state built as eq. (23). This is to be contrasted to the rotated basis representation used for the usual QISM, where an arbitrary set of $\tilde{\Lambda}_j$ will not, generically, correspond to an off-the-shell state since $M$ Bethe roots $\mu_k$ are insufficient to properly rebuild $N$ arbitrary values of $\Lambda_j$. This bijection between off-the-shell states and sets of $\Lambda_j$ is therefore exclusive to this representation.

## 7 Scalar products

### 7.1 Scalar products for a given field orientation

The proposed representation of Bethe states as defined in eq. (23) naturally allows for the construction of a simple determinant representation for the scalar product of an arbitrary off-the-shell state:

$$|\{\lambda_1 \dots \lambda_N\}\rangle \equiv \prod_{i=1}^{N} \left( B_0^+ + \sum_{k=1}^{N} \frac{S_k^+}{\lambda_i - \epsilon_k} \right) |\Omega\rangle. \tag{71}$$

and an arbitrary dual state built in the same fashion:

$$|\{\mu_1 \dots \mu_N\}\rangle \equiv \prod_{i=1}^{N} S^-(\mu_i) |\uparrow\uparrow \dots \uparrow\rangle \equiv \prod_{i=1}^{N} \left( B_0^- + \sum_{k=1}^{N} \frac{S_k^-}{\mu_i - \epsilon_k} \right) |\uparrow\uparrow \dots \uparrow\rangle. \tag{72}$$

Indeed, the resulting product is then simply given by:

$$\langle \{\mu_1 \dots \mu_N\}| \{\lambda_1 \dots \lambda_N\}\rangle = \langle \uparrow\uparrow \dots \uparrow| \prod_{i=1}^{2N} \left( B_0^+ + \sum_{k=1}^{N} \frac{S_k^+}{\nu_i - \epsilon_k} \right) |\downarrow\downarrow \dots \downarrow\rangle, \tag{73}$$

with $\{\nu_1 \dots \nu_{2N}\} = \{\lambda_1 \dots \lambda_N\} \cup \{\mu_1 \dots \mu_N\}$. This expression is only valid for a given common magnetic field orientation, i.e. $(B_0^-)^* = B_0^+$ and does not imply the possibility to write similar expressions for products between states built from differently oriented fields and therefore using different realisations of the quasiparticle creation operators $S^+(u)$.

For a given common orientation, this last equation only has non-zero contributions coming from the terms in the operator product which contain exactly one single copy of each of the $N$ local spin-raising operators. Any subset of $N$ Bethe roots taken out of the available $\{\nu_1 \dots \nu_{2N}\}$ can be associated, in every possible way, to one of the $N$ spins to be flipped. The other "unused" roots all contribute with their constant term $B_0^+$. This leads to the following explicit structure:

$$\langle \{\mu_1 \dots \mu_N\}| \{\lambda_1 \dots \lambda_N\}\rangle = (B_0^+)^N \sum_{L \in S^N} \left( \prod_{k=1}^{N} \frac{1}{\ell_k - \epsilon_k} \right), \tag{74}$$

where the sum is over $L = (\ell_1 \dots \ell_N)$ covering every possible permutation of every subset of cardinality N built out of the elements of the 2N-set $\{\nu_1 \dots \nu_{2N}\}$. For a given subset of roots $\mathfrak{L}^N \equiv \{\ell_1 \dots \ell_N\}$ (now an actual set and therefore no longer ordered), the sum over the possible permutations of this particular product is nothing but the permanent of the $N \times N$ Cauchy matrix (with elements $C_{i,j} = \frac{1}{\ell_i - \epsilon_j}$) built out of the two sets $\mathfrak{L}^N$ and $\{\epsilon_1 \dots \epsilon_N\}$:

$$\langle \{\mu_1 \dots \mu_N\}| \{\lambda_1 \dots \lambda_N\}\rangle = \left( B_0^+ \right)^N \sum_{\mathfrak{L}^N} \text{Perm}_N \, C_{\{\epsilon_1 \dots \epsilon_N\}}^{\mathfrak{L}^N}. \tag{75}$$

Using a proof similar to that found in [61], this full sum has been shown in [37] to be given by a simple determinant. We redirect the reader to [37] for the proof and simply state here the result:

$$\langle \{\mu_1 \dots \mu_N\}| \{\lambda_1 \dots \lambda_N\}\rangle = \left( B_0^+ \right)^N \, \text{Det}_N \, J_{\{\epsilon_1 \dots \epsilon_M\}}^{\{\nu_1 \dots \nu_{2N}\}}, \tag{76}$$

where the matrix elements are given by:

$$J_{aa} = \sum_{b \neq a}^{N} \frac{1}{\epsilon_a - \epsilon_b} - \sum_{p=1}^{2N} \frac{1}{\epsilon_a - \nu_p} = \sum_{b \neq a}^{N} \frac{1}{\epsilon_a - \epsilon_b} - \Lambda_a^\lambda - \Lambda_a^\mu,$$

$$J_{ab} = \frac{1}{\epsilon_a - \epsilon_b} \ \forall \ b \neq a, \tag{77}$$

with $\Lambda_a^\lambda \equiv \sum_{p=1}^N \frac{1}{\epsilon_a - \lambda_p}$, $\Lambda_a^\mu \equiv \sum_{p=1}^N \frac{1}{\epsilon_a - \mu_p}$, the respective $\Lambda$ variables built out the Bethe roots defining both states.

Let us mention again that this rewriting of the sum of Cauchy permanents into a single determinant is valid for arbitrary sets $\{\lambda_1 \dots \lambda_N\}$ and $\{\mu_1 \dots \mu_N\}$ and therefore this determinant expression is also valid when both states are off-the-shell: defined by arbitrary complex-valued $\lambda$s and $\mu$s without the restriction that they be solution to their respective Bethe equations.

## 7.2 Scalar products with canonical-basis states

Additionally, an advantage of this approach over the rotated basis ABA is that it gives a straightforward way to compute the scalar product between any eigenstate (and even off-the-shell Bethe states) and a canonical basis state defined by having each individual spin in either one of its local $S_i^z$ eigenstates $|\uparrow_i\rangle$ or $|\downarrow_i\rangle$. These states are the exact eigenstates of the system at infinitely large z-oriented magnetic field (or alternatively in the decoupled limit).

It is straightforward to see that such a scalar product with the canonical state in which spins $i_1, i_2 \dots i_M$ are up while the other ones are down is given by:

$$\langle \uparrow_{i_1} \dots \uparrow_{i_M} |\{\lambda_1 \dots \lambda_N\}\rangle = \left(B_0^+\right)^{N-M} \sum_{L \in S^M} \left(\prod_{k=1}^M \frac{1}{\ell_k - \epsilon_{i_k}}\right), \tag{78}$$

where the sum is, here again, over every permutation of every subset of cardinality M built out of the elements of the N-set $\{\lambda_1 \dots \lambda_N\}$. Just as in the preeceeding section, we have to pick, in every possible way, exactly $M$ Bethe roots out of the available $N$ ones and associate them, in any order with one of the spins to be flipped. The remaining unused roots contribute their constant part $B_0^+$ leading to the prefactor. Again, for any given subset of cardinality $M$ built out of $M$ given roots: $\mathfrak{L}^M \equiv \{\lambda_{i_1}, \lambda_{i_2} \dots \lambda_{i_M}\}$ the sum over permutations can be performed to find:

$$\langle \uparrow_{i_1} \dots \uparrow_{i_M} |\{\lambda_1 \dots \lambda_N\}\rangle = \left(B_0^+\right)^{N-M} \sum_{\mathfrak{L}^M} \text{Perm}_M \, C_{\{\epsilon_{i_1} \dots \epsilon_{i_k}\}}^{\mathfrak{L}^M}, \tag{79}$$

and the sum of Cauchy permanents is, again, equal to the single $M \times M$ determinant

$$\langle \uparrow_{i_1} \dots \uparrow_{i_M} |\{\lambda_1 \dots \lambda_N\}\rangle = \left(B_0^+\right)^{N-M} \, \text{Det}_M \, J_{\{\epsilon_{i_1} \dots \epsilon_{i_M}\}}^{\{\lambda_1 \dots \lambda_N\}}, \tag{80}$$

with matrix elements constructed as in (77) but using a restricted set of only $M$ spins:

$$J_{aa} = \sum_{b \neq a}^M \frac{1}{\epsilon_{i_a} - \epsilon_{i_b}} - \sum_{p=1}^N \frac{1}{\epsilon_{i_a} - \lambda_p} = \sum_{b \neq a}^M \frac{1}{\epsilon_{i_a} - \epsilon_{i_b}} - \Lambda_{i_a},$$

$$J_{ab} = \frac{1}{\epsilon_{i_a} - \epsilon_{i_b}} \ \forall \ b \neq a. \tag{81}$$

Let us mention again that this expression remains valid for arbitrary sets of $\{\lambda_1 \dots \lambda_N\}$. This simple projection formula would allow one to easily decompose any initial state of the form $|\uparrow_{i_1} \dots \uparrow_{i_M}\rangle$ on the true eigenbasis of the system, and to do so using exclusively the sets of eigenvalued-based $\Lambda_j$ variables. This, in turn, can allow efficient computation of the subsequent unitary time evolution induced by any integrable quantum Hamiltonian covered in this work.

# 8 Conclusion

By implementing the QISM on a "faulty" pseudovacuum, which is not a highest weight state and therefore not an eigenstate of the transfer matrix, we have built a common formalism which allows the construction of a Bethe ansatz for isotropic Richardson-Gaudin models in an arbitrarily oriented magnetic field. Each eigenstate is then characterised by a set of Bethe roots whose cardinality is always the same, namely that of the system size $N$. It also allows the construction of equivalent, albeit simpler to solve, quadratic Bethe equations which are written in terms of the conserved charges' eigenvalues.

In the isotropic XXX case treated here, the proposed formalism could be seen as superfluous since the rotation of the system's quantisation axis can already provide an approach to its exact Bethe ansatz solution. However, as we demonstrated, it does grant us easy access to a simple and useful determinant representation for the scalar product of eigenstates (and even off-shell Bethe states) and the common "canonical" basis states. This particular set of techniques can also provide a well-defined path to a full generalisation of these results to the XXZ case [38], where the anisotropy of the spin-spin coupling excludes the construction of a Bethe ansatz through a simple rotation of the quantisation axis.

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
