# Peer review of "Common framework and quadratic Bethe equations for rational Gaudin magnets in arbitrarily oriented magnetic fields"

_SciPost Physics, doi:SciPost Phys. 3, 009 (2017)_

## Round 1 · Referee Report · Anonymous (Referee 2) · 2017-4-26

Strengths

The authors solve analytically an integrable model using the algebraic Bethe ansatz but without pseudovacuum.

Weaknesses

They do not compare enough with the previous works on this subject.

Report

The authors diagonalise the XXX Gaudin model in an arbitrarily oriented exterior magnetic field with the algebraic Bethe ansatz (ABA). As they explained, by a rotation, this problem can be transformed to the usual one with the magnetic field in the z-direction and it can be solved by the usual ABA. However, they do not want to perform this transformation in order to solve all the models in a common framework. Therefore, they need to modify slightly the Bethe ansatz and must overcome some supplementary technical issues. This result seems interesting from the point of view of the methods used to solve it but also for the use in further computations as they discussed. This work deserves its publication in SciPost after the revision of the points given below.

Requested changes

1/ It would be interesting to compare the method used in this paper with the previous works concerning the algebraic Bethe ansatz without pseudovacuum. For example :
a/ the ansatz with the same number of the creation operators than the number of sites (see relation (22)) has been introduced previously in SIGMA 9 (2013), 072 ( arXiv:1309.6165);
b/ the computation of the supplementary term in their relation (26) which is the main technical problem has been also encountered in other contexts and solved by different methods in J. Phys. A48 (2015) 08FT01 and arXiv:1411.7954 ; N. Phys. B899 (2015) 229-246 and arXiv:1506.02147 ; SIGMA 11 (2015), 099 and arXiv:1506.06550.

2/ The type of Bethe equations they obtained has been introduced previously in the context of the spin chain without U(1) symmetry and has been the heart of a lot of recent researches initiated in Phys. Rev. Lett. 111, 137201 (2013) and arXiv:1305.7328v4.

3/ At the end of section 4.1, they discuss about the SoV used to solve the models without proper pseudovacuum. Other methods to deal with this problem have been also introduced. In addition of the references given previously (concerning the ABA and the TQ relation), there exist also the q-Onsager approach (J. Stat. Mech. (2007) P09006 and arXiv:hep-th/0703106), the generalisation of the coordinate Bethe ansatz (J. Stat. Mech. (2010) P11038 and arXiv:1009.4119) or the functional Bethe ansatz (J. Stat. Mech. (2005) P08002 and arXiv:hep-th/0504124; Nucl. Phys. B790 (2008) 524 and arXiv:0708.0009; J. Phys. A44 (2011) 015001 and arXiv:1009.1081) .

4/ I believe that the cases $B_0^+=0, B_0^- \neq 0$ and $B_0^+\neq 0, B_0^- = 0$ deserve at least a remark before to treat the general case with $B_0^+\neq 0, B_0^- \neq 0$. It would correspond to triangular boundaries treated for example by ABA in Lett. Math. Phys. 103 (2013) 493 and arXiv:1209.4269.

5/ To enlighten their results, I propose to reorganize a bit the section 4. The final results summarized at the end by relations (49)-(53) may be put after relations (22) and the rest of the section may be presented as a proof of these results.

6/ A discussion on the completeness of their solution is also necessary. As usual in the context of Bethe ansatz, they give a proof that the vector they proposed (assuming that it is not zero) is an eigenvector but we do not know a priori that all the eigenvectors are obtained by this way.

7/ I list below small typographic mistakes:
- in the last sentence of the first paragraph of the introduction: 'a eigenstates which are';
- at different places, they give the eigenvalues of Sz proportional to $\hbar$. If it is the case, $\hbar$ is also present in the commutation relations of the Gaudin algebra;
- they must precise that the epsilon are 2 by 2 different after (12);
- in (21), an index 'i' is missing for S;
- keep the same notations for the l.h.s of (5), (22), (50),...
- in (37) and (49), they used $B_0^z$ instead of $B_z$;
- a tilde is missing on the last $\Lambda$ of (64)

---

## Round 1 · Referee Report · Anonymous (Referee 3) · 2017-5-3

Strengths

1. Very well explained and readable paper
2. Very interesting result relating ABA-like and SOV-like constructions for a simple toy model
3. Connection between homogeneous and inhomogeneous Bethe roots

Weaknesses

1. Absence of a scalar product formula for off-shell and on-shell states (it seems that in this construction even a off-shell off-shell scalar products can be computable)

Report

The authors treat a simple toy model (Gaudin model in an arbitrary oriented magnetic field) but their approach can be quite useful for more physical situations. In the recent years a lot of attention was drawn to integrable spin system without simple reference state (such as spin chains with non-diagonal boundaries, spin chains with non diagonal boundary twists etc.). A particularly important question in this framework is a relation between the homogeneous and inhomogeneous Bethe equations. For this toy model the authors show clearly in a simple way how to proceed from the inhomogeneous equation (constructed in this paper) to the traditional homogeneous ones (which can be always obtained using the SU(2) symmetry of the model). Evidently this transformation is simple because the model considered is extremely simple. However there is a hope that this toy model can give some hints to understand more complicated situations.

The paper is well written, the statements are proved in a clear way. I recommend the paper for publication in SciPost.

Requested changes

1. Mention after eq. 12 that the inhomogeneity parameters \epsilon_j should be generic

---

## Round 1 · Referee Report · Anonymous (Referee 4) · 2017-5-12

Strengths

1- Alternative, possibly easier to generalize, solution to a model of Gaudin magnets in a magnetic field.
2- All statements are thoroughly proven.

Weaknesses

The method is only applied to a simple model so far.

Report

This paper deals with the XXX Gaudin model in an arbitrarily oriented magnetic field. The model may be solved by rotating the quantization axis, and then applying standard ABA techniques. The authors follow a different strategy, and their construction relies on a pseudovacuum which is the same for all magnetic fields. The price to pay is a slight adaptation of the ABA formalism, which is worked out in the present manuscript. The main hope is that the method presented here may also be applied to more complicated models where a simple rotation of the quantization axis does not work anymore, such as the XXZ case.

Overall the paper is quite interesting, and timely, given the recent interest in integrable systems without a simple reference state. The results are clearly stated and proven, which makes it easy to read also. For these reasons I recommend publication in SciPost, provided the minor issues mentioned below are addressed.

Requested changes

1- In section 4.1, the relation to SoV should be discussed more thoroughly.
2- What about the completeness of the set of eigenstates constructed here?
3- There is a clash of notations regarding the variable k in several equations, including (38),(39),(40),(41).
4- Section 5. While the derivation itself is clear, putting the main result (58) either at the beginning or at the end would definitely improve readability.
5- Last paragraph before the conclusion. Regarding the normalization of these eigenstate, the authors say either too little or too much.

Below is also a short list of additional typos:

a) The first sentence page 2, especially the "whose", reads awkwardly.
b) Second paragraph "leads" should read "leading".
b) Page 9 before (23). "writable" is improper in that context. Same comment on page 19.
c) In (71), the Cauchy matrix is strictly speaking not defined.
d) In the conclusion "and therefore an eigenstate" should read "and therefore not an eigenstate".

---

## Round 2 · Referee Report · Anonymous (Referee 5) · 2017-7-7

Strengths

The authors solve analytically an integrable model using the algebraic Bethe ansatz but without pseudovacuum.

Weaknesses

n/a

Report

The authors modified the paper as requested

Requested changes

No change

---

## Round 2 · Referee Report · Anonymous (Referee 6) · 2017-7-13

Strengths

Improved new version

Weaknesses

None

Report

The new version is improved with respect to the first one.

Requested changes

None

---

## Round 2 · Referee Report · Anonymous (Referee 7) · 2017-7-18

Strengths

Same as before

Weaknesses

Same as before

Report

I believe the minor issues raised by the referees have been all successfully addressed.

Requested changes

None

---

## Round 2 · List of Changes

The issue of completeness has been addressed in section 6. Indeed, in https://scipost.org/submissions/1603.03542v4/ the completeness of the Quadratic Bethe equations in the usual (z-oriented field) XXX case has been demonstrate. The simple one-one correspondence between eigenstates in the generic field and the z-oriented case which was presented in this section (simple shift of the eigenvalues) consequently leads to a “trivial” homotopy which preserves the completeness in the case treated here.

Following a referee’s suggestion, we have vastly expanded the discussion concerning other approaches (and consequently the list of references as well) which have been developed to deal with similar situations in a variety of integrable models. Instead of simply mentioning separation of variables, we present a broader picture of the many techniques, which should give the interested reader much more material to look into. From the sheer number of such approaches, it is in our opinion well beyond the scope of this work, to present explicitly the numerous formal possible links to our work.

We have also added references to Claeys’ work on the p+ip superconductor, which has also made use of the same technique in this explicit XXZ-Gaudin case as well as that of Lukayenko on the same subject which, despite not discussing explicitly the eigenstates of the system has also derived similar conserved charges, Bethe equations and eigenvalues.

Section 5 has been slightly modified to make the final result appear at the very end of the section while section 4 has been reworked in order to regroup the results at beginning and leave the rest of the section as proof.

We also added a short discussion of the “triangular boundary” case, where the proposed ansatz using as many Bethe roots as we have spins, would no longer be valid. In this case, eigenstates are defined by a smaller number of Bethe roots as the pseudovacuum is now a properly defined eigenstate of the transfer matrix.

As it was explicitly mentioned in one of the reports, we added subsection 7.1 to give explicit determinant formulas the scalar products between generic off-the-shell Bethe states and generic off-the-shell dual states, for a given fixed (but generic) orientation of the field. This has lead to a slight shortening of 7.2, since the formulas for the canonical basis projections are obtained in an more or less identical fashion as those in 7.1

The various typos, notational errors and small precisions requested in the various reports have also all been explicitly addressed.

---

## Editorial Decision

published